# Prevalence of Orthorexia in Groups of Students with Varied Diets and Physical Activity (Silesia, Poland)

**DOI:** 10.3390/nu14142816

**Published:** 2022-07-08

**Authors:** Mateusz Grajek, Karolina Krupa-Kotara, Krzysztof Sas-Nowosielski, Ewa Misterska, Joanna Kobza

**Affiliations:** 1Department of Public Health, Faculty of Health Sciences in Bytom, Medical University of Silesia in Katowice, 41902 Katowice, Poland; mgrajek@sum.edu.pl (M.G.); jkobza@sum.edu.pl (J.K.); 2Department of Humanistic Foundations of Physical Culture, Faculty of Physical Education, Jerzy Kukuczka Academy of Physical Education in Katowice, 40065 Katowice, Poland; gmattgrayek@gmail.com; 3Department of Epidemiology, Faculty of Health Sciences in Bytom, Medical University of Silesia in Katowice, 41902 Bytom, Poland; 4Department of Pedagogy and Psychology, Faculty of Social Studies in Poznań, Poznań School of Security, 60778 Poznań, Poland; emisterska1@wp.pl

**Keywords:** eating disorders, eating behaviors, students, nutrition, orthorexia, lifestyle, diet, physical activity, health-related, non-health-related

## Abstract

(1) Background: The literature emphasizes the role of many factors influencing the onset of eating disorders (EDs) and their mutual influence on each other. Therefore, this study aimed to evaluate and compare the prevalence of orthorexic behaviors in groups of health-related and non-health-related students in terms of their differential health behaviors—diet and physical activity levels. (2) The study included 300 individuals representing two equal groups of fields of study, which for the study were called the health-related field (HRF) and the non-health-related field (NRF). (3) Results: Based on the results of the dietary assessment, it was found that the best dietary model was characterized by the HRF group; in this group, 97.2% of students were characterized by a very good and good dietary mode. The NRF group, on the other hand, was dominated by a sufficient dietary mode for 64.4% of all cases in this group (94 people), while the dietary model marked as “good” was less popular, at 24.6% of this group (36 people). (4) Conclusions: Based on the cited self-research and information from the literature, it can be concluded that the problem of orthorexia is still a new issue at the level of social sciences, medical sciences, and health sciences. The psychometric tools used in this study allowed us to demonstrate the prevalence of the aforementioned eating disorders in the sample groups of students.

## 1. Introduction

Proper nutrition and adequate physical activity are the most important factors that promote health. However, sometimes, the interaction of certain factors leads to the manifestation of eating disorders (EDs). The literature emphasizes the role of many factors influencing the onset of eating disorders and their mutual influence on each other, the most frequently mentioned being physiological, psychological, and socio-cultural factors [1,2]. Eating disorders form a group of behavioral disorders that involve excessive weight control and focusing thoughts on food. According to the American Psychiatric Association’s classification (DSM-V), there are two primary groups of eating disorders—specific and nonspecific [3]. Specific eating disorders include anorexia (mental anorexia), bulimia (mental gluttony), and sometimes a mixed disorder (bulimorexia) [4]. Nonspecific eating disorders form a group of behaviors not classified in the DSM-V, but this does not mean that these individuals do not cause serious health consequences for the person who suffers from them [5]. Nonspecific eating disorders include pathological eating behaviors such as emotional eating or food fanaticism. According to the International Statistical Classification of Diseases and Health Disorders (ICD-10), eating disorders are classified as mental disorders from the group of behavioral disorders, associated with physiological disturbances and psychological factors. Eating disorders are classified as compulsive disorders due to the compulsion to restrict or overindulge in food. Among the most characteristic symptoms of these disorders are anxiety, depression, obsessive-compulsive disorder, self-aggression, or reduced quality of life [1,2]. Major risk factors for EDs include [6,7]:
-Physiological factors: female gender, abnormal functioning of neurotransmitters and neuromodulators (serotonin, dopamine, norepinephrine), genetic predisposition, abnormalities in hormonal axes (hypothalamic-pituitary).-Collective psychological factors: disturbed family and peer relations, the occurrence of psychological problems in the family (depression, anxiety disorders, obsessive-compulsive disorders), abnormal family patterns (dysfunctional family, overprotection from the mother or father, sexual abuse).-Individual psychological factors: distorted self-image, low self-esteem, perfectionism, intrapsychic and interpersonal communication difficulties, sense of responsibility, suppression of aggression, submissiveness.-Socio-cultural factors: the cult of a slim figure created by the media, social expectations of the role.

Namysłowska [8], on the other hand, indicates that factors influencing the development of eating disorders should be considered in terms of a causal sequence or vicious circle. The first group of factors mentioned by the author are predisposing factors, among which are the individual characteristics of the person at risk of developing the disorder, but also family and social issues (excessive ambitions, aggression, violence, social expectations). The second category of factors are triggered, and here behavioral disturbances within the desire for self-monitoring of body weight and the drive for self-acceptance are indicated. In the third category are the maintenance factors, which are defined by the effect of drastic dieting and physical activity in the form of starvation, which drive to maintain such a state of affairs—the person with an eating disorder feels satisfaction through achieving a given effect, and this causes the problem to worsen. Eating disorders are a significant public health problem. Their prevalence in the general population is quite high, despite difficult epidemiological studies. EDs are estimated to affect 2–4% of the population, and the prevalence of eating disorders is significantly higher in some age and social groups than in others [7,9]. Diagnosing eating disorders is a complex process, primarily due to the presence of nonspecific somatic symptoms that mask the underlying illness, thus prolonging the diagnosis process and making it more costly. In addition, the treatment of eating disorders is a lengthy and multi-step process that requires collaboration between the patient’s case manager, a psychologist or psychotherapist, and a nutritionist [2,4].

Nowadays, with the cult of slimness and care for a healthy lifestyle, there is a lot of talk about health risks associated with healthism, i.e., excessive attention to health. One can say that the eating disorder resulting from such behavior is orthorexia. Orthorexia nervosa is an informally diagnosed condition characterized by an obsessive preoccupation with eating healthy and “clean” food. In this condition, it is the quality, not the quantity, of food that is crucial [10,11]. Orthorexia nervosa, sometimes called radical eating, is a condition in which a pathological obsession with eating the “right” food is manifested. It is more commonly diagnosed in women and affluent individuals regardless of gender. Orthorexia affects 7% of the general population [1], and research by Dunn and Bartman [12] suggests that the problem may affect up to 90% of members of certain social groups, as the problem is more common in people with high levels of physical activity to maintain fitness and health, as well as in people who strive to achieve the “ideal” body shape that is promoted by mass media [13]. In this disorder, as in classic anorexia, there is an obsessive, systematic focus on counting the caloric value of the foods consumed and giving up all fatty and hard-to-digest foods [13,14]. Unlike anorexia, on the other hand, the focus of the sufferer is not only on the quantity consumed and weight reduction, but more importantly on the quality of the food consumed and on achieving and maintaining health [15,16]. Individuals with orthorexia think obsessively about food, planning meals, imposing dietary discipline on themselves, and enforcing punishments for breaking the rules they have established. Such behavior stems from the conviction that only eating healthy, easily digestible food and strict adherence to a diet will allow them to prevent the occurrence of diseases characteristic of modern societies-chronic non-communicable diseases [12]. Therefore, the desire to live a healthy and long life, to stay healthy and fit, and to focus on the quality of the food consumed rather than the quantity of the food consumed makes this disease not easy to diagnose [10,17]. There are also cases where patients who eliminate more products start consuming only water while believing that only drinking strictly selected water will ensure full health [5]. Health consequences of chronic orthorexia include, but are not limited to, broken social ties along with avoidance of going out to eat, malnutrition secondary to elimination and reduction diets (including deficiencies of protein, vitamin B12, iron, sodium), rhabdomyolysis (muscle breakdown), metabolic acidosis, and elevated laboratory indicators (mainly liver enzymes and bilirubin) [18]. Due to the lack of unambiguous diagnostic criteria, and thus the very great difficulty in making the diagnosis, the exact number of people with orthorexia is still unknown (apart from the aforementioned estimates, which are based on small group studies). People with orthorexia are often treated by their environment as caring too much about their health, and their relatives do not recognize the features of the disorder in such behavior. The lack of accurate information on the size of the condition is also due to the reluctance of sufferers to report to specialists and their failure to notice such problems [11,19]. Therapy for orthorexia, like therapy for anorexia, consists primarily of compensating for any nutritional deficiencies and psychotherapy. Its goal is to show patients that not all foods they reject carry health risks [12].

Therefore, the main objective of this study was to evaluate and compare the prevalence of orthorexic behaviors in groups of health-related and non-health-related students in terms of their differential health behaviors—diet and physical activity levels.

The following research hypotheses were posed in preparation for the study:Orthorexic behaviors are more common among people who have a rational diet.Orthorexic behaviors are more common among individuals who represent high levels of physical activity and motivation to engage in physical activity.Orthorexic behaviors are more common among people who overestimate the size and calorie portions of foods and foods.

## 2. Materials and Methods

### 2.1. Study Organization and Eligibility Criteria

The study included 300 individuals representing two equal groups of fields of study, which for the study were called the health-related field (HRF) and the non-health-related field (NRF).

The HRF group consisted of 150 final-year sophomore students with majors in dietetics and physical education. The key to selecting this group was the fact of having in-depth and professional knowledge in the field of rational nutrition and physical activity. The NRF group consisted of 150 students in their final year of second-degree studies with majors in management and computer science. The key to selecting this group was the fact that they did not have in-depth and professional knowledge in the field of rational nutrition and physical activity, at least at the university level. The assumption for the selection of these majors was that the gender groups were more or less equal. Such majors as dietetics and management are more often chosen by females, and physical education or computer science by males.

Individuals in the NRF group showing concurrent education (or past education) in a health-related field were excluded from the study. Individuals who had applied knowledge and skills in rational nutrition and physical activity in their professional work were treated similarly. The physiological state of the respondent was also taken into account. Persons suffering from diseases that influence the diet and/or physical activity of the respondent (e.g., allergies, food intolerances, metabolic diseases, tumors, etc.) were excluded from the research. The same was done with subjects who represented a specific dietary model (elimination diet or pregnancy and puerperium).

The final analyses included 290 (144 subjects in the HRF group and 146 subjects in the NRF group) correctly completed questionnaires that addressed the inclusion criteria described above. All participants were informed about the purpose and scope of the study and gave informed consent to participate.

The study was approved by the Bioethics Committee of the Medical University of Silesia in Katowice, in light of the Act on Medical and Dental Professions of 5 December 1996, which includes a definition of medical experimentation. The study participants consciously agreed to participate in the study.

### 2.2. Study Procedure and Research Tool

A diagnostic survey method was used in this study. The survey was conducted using an online form, which is an acceptable method in psychological research. The research was conducted using a survey technique. The survey consisted of a survey questionnaire and a scrapbook of sample foods and dishes. To record information from the research, a research tool was used, which included the following:I.The questionnaire of the survey consisted of a metric (data of the subject: gender, age, a field of study and occupation, anthropometric data—declared height and body weight); the author’s questionnaire of dietary habits based on the guidelines and standards of the National Institute of Public Health and the NCEZ [18,20]; questions about the physical activity practiced and its level based on WHO guidelines [21]; standardized questionnaires such as the Exercise Motivation Inventory (EMI-2) (validated by Sas-Nowosielski and Nowicka [22]), the Orthorexia Questionnaire (ORTO-15) (validated by Stochel et al. [23]; Brytek-Matera [24]). The survey questionnaire was available online from May–June 2021.II.An album of sample foods and dishes to verify the ability to estimate the size and calorie content of portions, consisting of 12 photographs consistent with the division of foods into 12 groups (one photograph per group) [25,26]. The study using the album was conducted with the sensory panel of the Department of Dietetics, Faculty of Health Sciences in Bytom, Silesian Medical University in Katowice, Poland, from July–August 2021. Before each study, visual perception (perception of images) was tested using a scrapbook. For this purpose, selected Ishihara boards and optical illusion boards were used. Both tools are commonly used to assess so-called visual daltonism and the perception of objects in pictures (e.g., size, shape, length). To link the results of the questionnaire with the album, each participant of the study was given an individual number while filling in the questionnaire, which was then also entered in the album. The study was conducted according to scientific ethics, anonymity rules, and the RODO clause (Polish Law on Respect for Classified Information).

### 2.3. Interpretation of the Tools Used

Body mass index was calculated using the formula: BMI [kg/m^2^] = body weight [kg]/height [m]^2^. The results were then interpreted using a scale [27]: >30 kg/m^2^—obesity (alarming score); 25–29.9 kg/m^2^—overweight (elevated body weight); 20–24.9 kg/m^2^—normal body weight; 17–19.9 kg/m^2^—underweight (underweight); <17 kg/m^2^—malnutrition (alarming score).

In the assessment of dietary intake, the author’s tool based on nutrition standards for the Polish population [18,27] was used, which included 20 dietary indices (e.g., frequency of consumption of individual product groups, number of meals during the day, regularity of meals during the day, snacking, fluids drunk). One point was awarded for each correct answer (in accordance with the applied standards), so the highest possible total score was 20. To prioritize the results, the following scale was adopted: 18–20 points—very good nutrition; 14–17 pts.—good; 10–13 pts.—moderate, ≤9 pts.—poor nutrition (bad).

Based on the physical activity score in the questionnaire, respondents were assigned a physical activity index (PAL), which takes the following values, based on current recommendations for physical activity, respectively [21,28]: 1.2—no physical activity; 1.4—low physical activity (approximately 140 min per week); 1.6—medium physical activity (approximately 280 min per week); 1.8—high physical activity (approximately 420 min per week); 2.0—very high physical activity (approximately 560 min per week).

The EMI-2 questionnaire consists of 56 variables grouped into 14 categories corresponding to motives for exercising. In each subscale, the respondent could receive 5 points, where 0 means the highest priority for the motivator and 5 the lowest; the lower the scores for a given motivator, the higher the motivation [29].

The ORTO-15 questionnaire consists of 15 questions, each to be answered on a 4-point scale (always, often, rarely, or never). Answers indicating a tendency toward orthorexic behavior receive 1 point, while those corresponding to normal eating receive 4 points. According to the creators, the cutoff point is a score of 40 points; scoring below this score indicates a tendency toward orthorexia [13].

### 2.4. Statistical Compilation

Tables and figures were prepared for all extracted data from the survey questionnaire and descriptive statistics (percentages (%), counts (N; n), mean (X) or median (M)values, standard deviations (SD), minimum and maximum values (MIN and MAX)) were included. Detailed statistical analyses were conducted, regarding the demonstration of differences between the represented behaviors (pro-health or anti-health) and the occurrence of orthorexia in the sample group. To analyze the above material, the χ^2^ (chi-square) test for nonparametric variables and the V-Cramér coefficient of the strength of the relationship (with Yates and Fisher’s correction) were used, as well as the nonparametric equivalent of the one-way analysis of variance, the Kruskal–Wallis test with the coefficient ε^2^. A probability level of *p* = 0.05 was assumed for the study.

## 3. Results

In terms of gender distribution, the sample groups were as follows: women, 60.0%—174 persons (HRF: 47.1%, n = 82; NRF: 52.9%, n = 92); men, 40.0%—116 persons (HRF: 53.4%, n = 62; NRF: 46.6%, n = 54). All respondents were students in the final year of their master’s degree (second year of their second degree). Based on the medical history, it was observed that 5.2% (15 persons) had been diagnosed with chronic diseases, although these were seasonal allergies (pollen, house dust mite, insect venom), i.e., diseases that do not significantly affect their lifestyle. The age of the respondents was 26 years (±2 years). Dietetics was studied by 48.6% of subjects (n = 70) and physical education by 51.4% of subjects (n = 72)—these subjects constituted the HRF group (144 subjects). Management was studied by 47.3% of individuals (n = 69) and computer science by 52.7% (n = 77)—these individuals constituted the NRF group (146 individuals). Among the respondents, 249 people (85.9%) lived in large cities (over 100,000 inhabitants), 23 people (7.9%) lived in smaller cities (under 100,000 inhabitants), and 18 people (6.2%) lived in villages. As far as their occupation was concerned, only 13.1% of the respondents (38 persons) had permanent employment, i.e., in telecommunication, service, and administration-office sectors. The main addiction in the surveyed groups was smoking tobacco, with 3.8% of students (11 persons) admitting to this habit. There were no subjects who compulsively consumed alcohol or took other psychoactive drugs.

None of the subjects in the study were characterized as malnourished. More than 15.2% of the subjects were characterized as underweight (44 subjects in the HRF group). Normal weight was a characteristic of 178 subjects (61.3%). Overweight and obesity were present only in the NRF group with a total of 68 subjects (23.4%). These differences were statistically significant (T = 11.281; V = 0.621, *p* = 0.001). The results of the calculations are presented in Table 1.

Based on the results of the dietary assessment, it was found that the best dietary model was characterized by the HRF group; in this group, 97.2% of students were characterized by a very good and good dietary mode (84.0%—121 persons, 13.4%—19 persons, respectively). The NRF group, on the other hand, was dominated by sufficient dietary mode, at 64.4% of all cases in this group (94 people). Less popular was the dietary model marked as “good”, with only 24.6% of this group (36 people). It should be emphasized that an incorrect dietary pattern was represented only by people from the HRF group (3.9% of the total number of subjects—11 people). Detailed results of the students’ dietary assessment are shown in Figure 1.

The next stage of the study was to assess the physical activity and motivation of the respondents to undertake physical activity. The level of physical activity was assessed by answering four questions regarding the fact of exercise, type of exercise, frequency, and duration of physical activity. Based on the data obtained, it was observed that 98.6% (142 persons) of the HRF group and 83.6% (122 persons) of the NRF group were physically active. Correspondingly, 2 individuals from the HRF and 24 individuals from the NRF did not engage in any physical exercise daily (1.4% vs. 16.4%)—these individuals were not included in Figure 2 which categorizes physical activity levels by PAL index.

Low physical activity was characteristic for 46.2% of respondents (122 persons), and most often chosen by persons from the NRF group (79.5%—97 persons). Medium physical activity was observed in 25.7% of the respondents (68 persons); this activity concerned both the HRF group (33.8%—48 persons) and the NRF group (16.4%—20 persons). Physical activity at a high and very high level concerned 28.1% of the students (75 persons), these were mainly persons from the HRF group (48.4%—70 persons) (Figure 2).

The level of motivation to undertake physical exercise and its reasons in the sample group varied, and were assessed using the EMI-2 questionnaire. In the sample group, the pleasure derived from physical activity was the most important motivation to undertake it (1.79 ± 1.78). Psychological regeneration (1.81 ± 1.73), maintaining health (1.91 ± 1.82), building strength and endurance (1.93 ± 1.81), taking care of appearance (1.96 ± 1.86), and avoiding ill health (1.99 ± 1.71) were also strong motivators. The least important motivation for exercise was health pressure (2.77 ± 2.22), followed by social recognition (2.62 ± 1.93) and the desire to belong to a group (2.37 ± 2.02). There were no statistically significant differences between groups. The results were comparable in both groups of students (Table 2).

Before evaluating the size and caloric value of the portions using the albums of exemplary products and dishes, each participant had to pass a visual perception test consisting of the evaluation of six images divided into two groups: (1) Ishihara boards—assessing the perception of colors, and representing in turn: the number 12, the number 96 and the absence of a specific shape; and (2) optical illusion boards—assessing the perception of space (size, length, and shape). Each correct indication in the visual perception test was scored one point, and a total of six points could be gained. In the sample group, 89.6% (n = 282) scored from five to six points, and 10.4% (n = 30) scored three to four points. The greatest difficulty for the subjects turned out to be the Ishihara board, which does not represent a specific shape, and the optical illusion board, which assesses shape perception (assessing the straightness of the lines depicted). Due to the overall good performance on the test, it was not decided to exclude any participant in the remainder of the study.

The visual perception assessment was followed by the actual test of estimating portion sizes and calories. Each participant received individual albums of sample products and dishes, which contained 12 photographs of food products and ready meals. The evaluator’s task was to estimate the size and caloric value of the given item using the variants listed under the photographs (four variants for size (g), and four for energy supply (kcal)). The variants were selected so that one indicated the correct value; one smaller or larger in the range of 8–12% from the correct value, but still treated as a correct answer; and two incorrect variants, underestimated and overestimated in the range of 28–32% relative to the correct value. Based on the data obtained, it was determined what percentage of respondents correctly and incorrectly (over- or underestimated) the size and calorie content of the servings.

Figure 3 shows the results of the portion size estimation test.

Based on the test on the ability to estimate the size of the portions based on photographs, it was found that 32.4% (94 people) overestimated the size of the portions of products and dishes indicated in the photographs. In this group, there were mostly people studying in health faculties—57.6% (83 people); less often, there are people from other faculties—7.5% (11 people). In the case of underestimation (33.8%, n = 98), the situation was reversed—people from the NRF group mainly underestimated the size of products and dishes—56.2% (n = 82); in the HRF group, much fewer people underestimated (11.1%, n = 16). The remaining persons correctly indicated the size of the portion—33.8% (98 people).

Figure 4 shows the results of the caloric estimation test for portions.

Analyzing the results of the test on the ability to estimate the calorie content of portions based on photographs, it was observed that 35.8% (104 people) overestimated the calorie content of the products and dishes indicated in the photographs. This group included mainly health-related people (58.3%—84 people), and less frequently, non-health-related people (13.0%—19 people). On the other hand, in the case of underestimation of the energy of dishes (35.2%, n = 102), people from the NRF group mostly underestimated the caloric value of products and dishes presented in the album (55.5%, n = 81); in the HRF group, such cases were much less (15.3%, n = 22). The remaining persons correctly indicated the calorie content of the portion—29.0% (84).

Using the ORTO-15 questionnaire, it was found that, among the subjects, 44.5% (129 subjects) scored below 40, indicating an increased risk of orthorexia. This was more frequent in the HRF group than in the NRF group (63.5% vs. 25.8%), with 92 and 37 subjects, respectively, from the student groups. Based on statistical inference, it was found that low BMI values occur in individuals from the HRF group, which indicates the presence of a statistically significant relationship between the indicated characteristics (T = 13.238; V = 0.723; *p* = 0.0001). Similarly, also in the case of diet, individuals representing good (27.6%) and very good (16.2%) nutrition were more likely to belong to the group of people with an increased risk of orthorexia. In this case, as well, a statistically significant correlation with HRF group membership was shown (T = 10.984; V = 0.683; *p* = 0.0001) (Table 3).

Next, it was decided to verify the relationships between the occurrence of orthorexic behavior and the represented level of physical activity. Based on the statistical inference performed, it was found that high PAL values occur in the HRF group, which indicates the presence of a statistically significant relationship between the indicated characteristics (T = 8.117; V = 0.597; *p* = 0.002) (Table 4).

The last verification concerned the relationship between the occurrence of orthorexic behaviors in the sample group and the ability of the respondents to estimate portion sizes and calories. Based on the results presented in Table 5 and the statistical analyses performed, it was concluded that there is a statistical relationship both in the case of estimation of portion size and caloricity of meals; individuals from the HRF group characterized by orthorexic behavior are more likely to overestimate the size and caloricity of the meal (*p* < 0.05).

## 4. Discussion

Several studies indicate that an occupational group with an increased risk of orthorexic behaviors are people who practice sports (representing various sports disciplines, including those that require a very slim and limber body). Due to the extreme difficulty in making a definitive diagnosis, the exact number of people with orthorexia is still unknown. Dunn and Bartman [12] estimate that between 6 and 90% of people may be affected depending on their social group (this spectrum is dictated by the different prevalence of the condition in different populations). People with orthorexia are very often treated by their environment as caring too much about their health, and their loved ones do not find features of the disorder in such behavior. The lack of accurate information on the size of the condition is also due to the reluctance of affected individuals to report to specialists and their failure to notice such problems.

Although orthorexia is not commonly recognized as an eating disorder, there is ongoing research on the condition of overeating, limiting food intake, and leading a pathologically pro-healthy lifestyle. The following is a review of studies that look at the relationship between psychological state, occupation/field of study, practicing physical activity, or anthropometric indices, and the occurrence of orthorexic behavior. In recent years, there have been several papers regarding the condition of orthorexia. Current scientific evidence is presented below and compared with the results of our research.

In our own study, the occurrence of orthorexia depended on the field of study chosen by the participants of the study. Of course, the authors do not suggest that it is the field of study that influences the incidence of orthorexia, but rather the personality conditions that determine its choice, which also influence the development of orthorexic behavior. In the current study, it was observed that orthorexia occurs twice as often in people from the health-related group than in non-health-related people (63.5% vs. 25.8%). In addition, Kinzl et al. [30] in their study checked the prevalence of orthorexia among 283 Austrian female dietitians. They used the German version of the FEV questionnaire, which examines three dimensions of eating behavior, and the ORTO-15 questionnaire to verify the presence of orthorexia. In the sample group, more than one-third were individuals who had recently changed their eating habits. Among the respondents, some orthorexic behavior was presented in 34.9% of people, while orthorexia was found in 12.8% of people. Individuals in the last group were more likely than others to have experienced eating disorders recently. From the FEV questionnaire, it was found that 40% of individuals significantly controlled their eating behavior. A study by Haddad et al. [31] analyzed a group of 811 Lebanese people, from each province. The paper also used the ORTO-15 questionnaire and adopted the same cut-off point as in our study. The study found that 75.2% of the respondents exhibited tendencies and behaviors indicative of orthorexia. There was a significant association of orthorexia in the group of women, with people starving themselves to lose weight, urging others to “go on a diet”, and claiming that eating out is unhealthy.

The current study confirmed that people with higher physical activity are more likely to engage in orthorexic behavior. The demonstrated dependence showed that the higher the PAL index, the higher the risk of orthorexic behaviors. Moreover, the PAL index at the level of 2.0 was present only in the group of people with a positive ORTO-15 result. Malmborg et al. [32] investigated health status, physical activity, and frequency of orthorexia among physical education and management students. Respondents completed the Short Form Quality of Life Assessment Questionnaire (SF-36), the International Physical Activity Questionnaire (IPAQ), and the ORTO-15 questionnaire. Of 188 students, 144 (76.6%) had an ORTO-15 score indicating orthorexia, of which 84.5% were physical education students. Orthorexia combined with high levels of physical activity was more commonly observed in male physical education students than in female physical education students (45.1% vs. 8.3%). This would support the hypothesis posed in our study—those who participate intensively in sports may have a significantly higher exposure to orthorexia than those with lower levels of physical activity. This hypothesis is also supported by other research conducted by the author of the present study; in the 2020 study, the risk of orthorexia estimated based on ORTO-15 was determined for three-quarters of the sample group (300 out of 400 ballet schoolgirls in Silesian province) [33].

In addition, in our own study, it was shown that there is a true relationship, which indicates that orthorexia may occur in the group of people with a low BMI index; however, it was first of all noticed that it is appropriate for the group of people with normal body weight, regardless of the field of study they represented (HRF vs. NRF–16.6% vs. 12.8%). Similarly, Agopyan et al. [34] conducted a study to determine the relationship between orthorexia and the body composition of female students at Marmara University. The participants of the study were female students whose scores on the ORTO-15 questionnaire and EAT-40 eating attitude test indicated the presence of orthorexia. Evaluation of the respondents’ body composition indices (bioelectrical impedance on a Tanita SC-330 device) showed that there was no significant difference between the EAT-40 and ORTO-15 scores in terms of body composition. The vast majority of female respondents (70.6%) had high ORTO-15 scores, and there was a significant negative correlation (*p* < 0.05) between EAT-40 and ORTO-15 scores. The final results of the data analysis showed that although abnormal tendencies were common among the female students, they were able to maintain normal body composition. Similarly, a study by Grammatikopoulou et al. [35] showed that health sciences students, especially nutrition and dietetics students, have a higher prevalence of eating disorders. A total of 176 undergraduate students from the Faculty of Nutrition and Dietetics in Greece participated in the study. The study monitored food intake and examined the frequency of eating as a result of emotion and stress (EADES). Among the students participating in the study, 4.5% showed food addiction, and 68.2% showed orthorexia. No differences were observed between males and females in food addiction and orthorexia. Students with orthorexia showed increased BMI. Orthorexics consumed more low-energy foods, including vegetables, and less high-energy foods (meat and fat). Multiple linear regression analysis showed that orthorexic behavior was associated with increased BMI, waist circumference, and energy intake. Lower BMI was associated with the ability to cope with EADES. 

Farchakh et al. [36] conducted a study to assess the association between orthorexic behaviors (ORTO-15), eating habits (EAT-40), and anxiety levels (HAM-A) among a representative sample of medical students in Lebanese universities. A total of 627 medical students participated in the study. Linear regression results showed that a higher EAT-40 questionnaire score was significantly associated with lower ORTO-15 scores (more orthorexic behaviors), while a higher HAM-A anxiety score was significantly associated with higher ORTO-15 scores (less orthorexic behaviors). The results obtained in this study suggest an association between eating habits indicative of an eating disorder and orthorexic behaviors. In addition, individuals with orthorexia were shown to be less likely to experience anxiety. The self-reported study also highlighted the fact that there is an association between eating patterns and the occurrence of specific disorders. Barnes and Caltabiano [37] examined the correlations between perfectionism, body image, attachment style, self-esteem, and the occurrence of orthorexia. In total, 220 subjects completed a questionnaire consisting of the ORTO-15 questionnaire, the Multidimensional Perfectionism Scale (MPS), the Multidimensional Body Image Questionnaire (MBSRQ-AS), the Relationship Scale Questionnaire (RSQ), and the Rosenberg Self-Esteem Scale (RSES). The study confirmed a significant association of orthorexia prevalence among individuals with scores indicating perfectionism, appearance orientation, preoccupation with overweight and self-esteem, and aversion to attachment. However, the association of orthorexia prevalence with self-esteem was not confirmed. In our study, the occurrence of orthorexia nervosa was not compared with the mental state of the respondents, but an important correlation was shown that people suffering from this disorder more often overestimate the energy and volume value of food portions, which may indicate that among these people, similarly to other EDs, there are cognitive disorders associated with the incorrect perception of not only your own body, but also food intended for consumption. This condition should be the subject of further research on this topic.

### Strengths and Limitations

The research on the prevalence of orthorexic behaviors among students of different majors allowed us to understand the basic mechanisms, cause–effect relationships, and determinants of the occurrence of the indicated disorders. Conducting the research required a lot of work and preparation in the form of developing research tools and becoming familiar with existing psychometric tools measuring the risk of orthorexia. Of course, the paper does not suggest that it is the field of study that influences the development of the disorder, but rather that individuals who choose it are characterized by certain traits that predispose them to it. This should be understood in the way that, thanks to the results of the study, it is possible to detect groups of people who should be included in the observation in terms of the control and safety of their lifestyle.

An important limitation of the study is that causation cannot be described as it is a cross-sectional study. The main difficulties during the conduct of the study were access to the student group because the study was conducted in the period of May–June 2021, and it should be emphasized that this was the period immediately adjacent to the lifting of the COVID-19 pandemic restrictions, so it was necessary to use the method of online surveying using Google package forms. In addition, data such as height or weight, because the study was conducted using the CAWI method, were provided by the respondents themselves, which may result in a mistake caused by the self-assessment of these indicators by the participants. The use of an Internet survey still raises some doubts among researchers as to the reliability of filling out the questionnaires. It is worth emphasizing here that despite many reservations about online surveys, it is currently one of the main and most acceptable forms of conducting them. The second part of the research, which involved the use of food and dishes photo albums, was conducted face-to-face. This approach was chosen because of the extensive nature of the research tools and concerns about respondents’ reluctance to fill out the extensive questionnaire. In addition, contemporary reports confirm that people are more willing to participate in research when the researcher provides a comfortable research experience (online methods, neutral environment, interesting setting) [38]. The research material presented in this study, in light of scientific reports, appears to be a rather novel approach to the topic of nonspecific eating disorders, as research rarely uses methods from psychology and dietetics simultaneously. This approach has allowed for a comprehensive analysis and elaboration of the collected research material.

## 5. Conclusions

Based on the cited self-research and information from the literature, it can be concluded that the problem of orthorexia is still a new issue at the level of social sciences, medical sciences, and health sciences. The mechanisms of development of these disorders are still poorly understood, and any data resulting from the study are only a signpost for further scientific investigation. The psychometric tools used in this study allowed us to demonstrate the prevalence of the aforementioned eating disorders in the studied groups of students, along with their determinants, and the photo album of selected foods and dishes allowed us to estimate the respondents’ skills in estimating the weight and energy of food intake. Nevertheless, it is worth noting that the methods used were characterized by certain limitations; therefore, in the future, it would be worth extending the scope of the study to other social groups, a wider range of research tools, and deepening the existing ones.

The conducted research allowed us to verify the set research hypotheses and to formulate the following conclusions: orthorexic behaviors are characteristic of a group of people who are associated with health, and who take care of their lifestyle, diet, and participate in physical activity. Individuals who are at increased risk for orthorexia tend to overestimate the size and caloric content of their meals. The dietary condition discussed in the paper should be under constant control, and all health promotion activities should focus on the nutrition education and psychoeducation of young adults.

In conclusion, it should be noted that a necessary aspect of further research on the topic of nonspecific eating disorders should be the development of actions aimed at the prevention of eating disorder development in young adults. This can be achieved by including mental problems of nutritional nature in the National Health Program and the National Mental Health Program, or by planning and implementing nationwide health policy programs that include such tasks as comprehensive health education, with particular emphasis on nutritional education and psychoeducation from the early school years. It should be noted that eating disorders are an important multidisciplinary problem. Apart from the well-known disorders, the less-known ones, whose background may be found in the trend for a “fit life”, are more frequently discussed. Therefore, the mentioned education should be provided not only by psychologists, but also by pedagogues, dieticians, personal trainers, and representatives of other professions connected with the problem.

## Figures and Tables

**Figure 1 nutrients-14-02816-f001:**
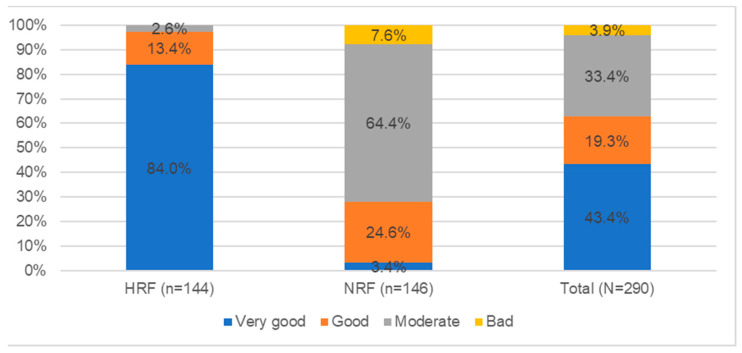
Students’ dietary assessment (N = 290): HRF—health-related field students; NRF—non-health-related field.

**Figure 2 nutrients-14-02816-f002:**
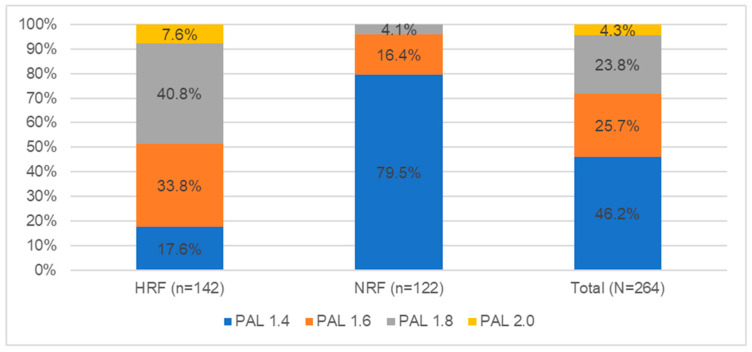
Students’ physical activity level ratings (N = 264).

**Figure 3 nutrients-14-02816-f003:**
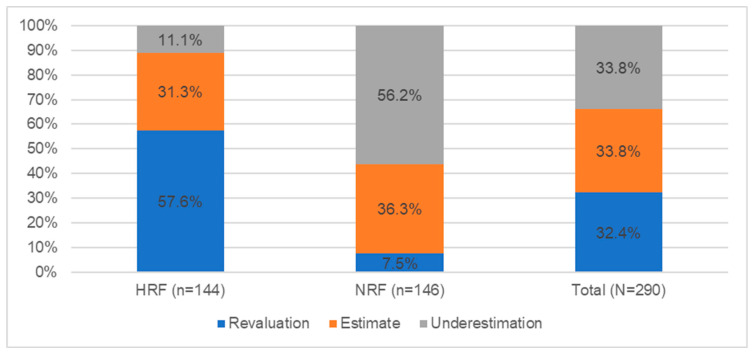
Level of portion size estimation across student groups (N = 290).

**Figure 4 nutrients-14-02816-f004:**
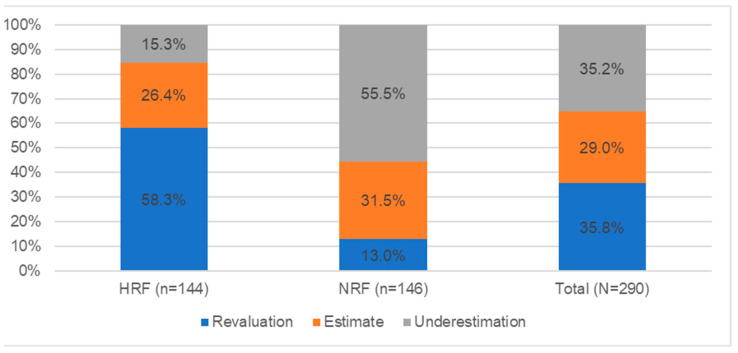
Level of calorie estimation of servings in student groups (N = 290).

**Table 1 nutrients-14-02816-t001:** Body mass indexes (BMI) of the students (N = 290).

Group	Malnutrition	Underweight	Normoweight	Overweight	Obesity
**HRF** **(n = 144)**	0 (0.0%)	44 (30.6%)	100 (69.4%)	0 (0.0%)	0 (0.0%)
**NRF** **(n = 146)**	0 (0.0%)	0 (0.0%)	78 (54.4%)	61 (41.7%)	7 (3.9%)
**Total** **(N = 290)**	0 (0.0%)	44 (15.2%)	178 (61.3%)	61 (21.0%)	7 (2.4%)
**T**	-		-	**11.281**	
** *p* **	>0.05		>0.05	**0.0001**	

HRF—health-related field students; NRF—non-health-related field. Bold values are statistically significant.

**Table 2 nutrients-14-02816-t002:** EMI-2 subscales by group (N = 290).

Subscale	Mean (X ± SD)			T	*p*
	HRF (n = 144)	NRF (n = 146)	Total (N = 290)		
**Stress Management**	2.20 (±1.32)	1.80 (±2.02)	2.00 (±1.72)	-	>0.05
**Mental Regeneration**	1.82 (±1.74)	1.79 (±1.72)	1.81 (±1.73)		
**Pleasure**	1.80 (±1.79)	1.78 (±1.77)	1.79 (±1.78)		
**Challenges**	2.04 (±1.81)	2.06 (±1.83)	2.05 (±1.82)		
**Social Recognition**	2.63 (±1.92)	2.61 (±1.94)	2.62 (±1.93)		
**Group Membership**	2.36 (±2.01)	2.38 (±2.03)	2.37 (±2.02)		
**Competition**	2.32 (±2.04)	2.35 (±2.06)	2.34 (±2.05)		
**Health Pressure**	2.78 (±2.23)	2.76 (±2.21)	2.77 (±2.22)		
**Avoiding Ill Health**	2.00 (±1.74)	1.98 (±1.68)	1.99 (±1.71)		
**Staying Healthy**	1.95 (±1.83)	1.86 (±1.81)	1.91 (±1.82)		
**Weight Control**	2.03 (±1.89)	2.09 (±1.95)	2.06 (±1.92)		
**Appearance**	1.90 (±1.80)	2.02 (±1.92)	1.96 (±1.86)		
**Strength and Endurance**	1.90 (±1.84)	1.96 (±1.78)	1.93 (±1.81)		
**Agility and Flexibility**	2.00 (±1.70)	2.18 (±1.72)	2.09 (±1.71)		

**Table 3 nutrients-14-02816-t003:** Relationship between BMI and diet and the group at increased risk for orthorexia (n = 129).

**BMI vs.** **ORTO-15**	**Malnutrition**	**Underweight**	**Normoweight**	**Overweight**	**Obesity**	**T**	** *p* **
	0	44 (15.2%)	85 (29.3%)	0	0	**13.238**	**0.0001**
**HRF (n = 92)**	0	44 (15.2%)	48 (16.6%)	0	0		
**NRF (n = 37)**	0	0	37 (12.8%)	0	0		
**Diet vs.** **ORTO-15**	**Bad**	**Moderate**	**Good**	**Very Good**		**T**	** *p* **
	0	2 (0.7%)	80 (27.6%)	47 (16.2%)		**10.984**	**0.0001**
**HRF (n = 92)**	0	0	45 (15.5%)	47 (16.2%)			
**NRF (n = 37)**	0	2 (0.7%)	35 (12.1%)	0			

Bold values are statistically significant.

**Table 4 nutrients-14-02816-t004:** Relationship between physical activity level and the group at increased risk for orthorexia (n = 129).

**PAL vs. ORTO-15**	**PAL 1.4**	**PAL 1.6**	**PAL 1.8**	**PAL 2.0**	**T**	** *p* **
	19 (6.6%)	27 (9.3%)	51 (17.6%)	32 (11.0%)	**8.117**	**0.002**
**HRF (n = 92)**	10 (3.4%)	15 (5.2%)	35 (12.1%)	32 (11.0%)		
**NRF (n = 37)**	9 (3.1%)	12 (4.1%)	16 (5.5%)	0		

Bold values are statistically significant.

**Table 5 nutrients-14-02816-t005:** Relationship between demonstrated ability to estimate portion size and group at increased risk for orthorexia (n = 129).

Estimating Portion Size vs. ORTO-15/TFEQ-13	Underestimation	Estimate	Revaluation	T	*p*
**Orthorexic Behavior (n = 129)**	0	32 (11.0%)	95 (32.8%)	**12.467**	**0.0001**
**HRF (n = 92)**	0	12 (4.1%)	78 (26.9%)		
**NRF (n = 37)**	0	20 (6.9%)	17 (5.9%)		
**Estimating Caloric Intake vs. ORTO-15**	**Underestimation**	**Estimate**	**Revaluation**	**T**	** *p* **
**Orthorexic Behavior (n = 129)**	18 (6.2%)	22 (7.6%)	87 (30.0%)	**11.551**	**0.0001**
**HRF (n = 92)**	8 (2.8%)	12 (4.1%)	70 (24.1%)		
**NRF (n = 37)**	10 (3.4%)	10 (3.4%)	17 (5.9%)		

Bold values are statistically significant.

## Data Availability

The original contributions presented in the study are included in the article; further inquiries can be directed to the corresponding author.

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
