# Peer review of "Prevalence of Orthorexia in Groups of Students with Varied Diets and Physical Activity (Silesia, Poland)"

_nutrients, 2022, doi:10.3390/nu14142816_

Round 1

Reviewer 1 Report

The research topic on orthorexia is interesting

Authors could use data from the literature review on orthorexia published in 2021: https:// doi.org/10.3390/ijerph18105488

Section method:  the characteristic of the study group show results and should be in the section results

Tool based on popular principles of rational nutrition  should be developped (Jarosz et al., 2020) line 220
 L225 duration of physical activity, should also be specified ?
It is not relevant to show the results of the total sample because the balanced sample between the 2 group is chosen for the study

Discussion :this section does not discuss the results of the study but is a review of the literature

Minor revisions:

one number after the decimal point in the figures

Figure 7 l 170 is not represented

Author Response

Dear Reviewer,

thank you for the suggested corrections to the text. We are also grateful for catching any omissions that we did not avoid in editing the manuscript. All changes indicated in the review have been applied and marked in red in the new version of the manuscript. We used the recommended literature sources in the introduction, improved the presentation of data in charts, tables and descriptions, expanded the discussion of the work with our own results and cited the limitations of our study at the end of the article (before the conclusions). The text has also been linguistically improved and any terminological inconsistencies have been removed. We believe that the article in its current form deserves to be published in the journal.

Thank you once again.

Best regards,
Authors

Reviewer 2 Report

Dear authors this is a very interesting study. Methods and sample are presented in detail and you have performed important tests.

However, there are some points that need improvement:

Introduction: While it presents sufficient data on eating disorders and orthorexia, you could provide more information particularly on orthorexia (please notice a 2022 review: Gkiouleka, M.; Stavraki, C.; Sergentanis, T.N.; et al. Orthorexia Nervosa in Adolescents and Young Adults: A Literature Review. Children 2022, 9, 365. https://doi.org/10.3390/ children9030365).

Results: Tables 1 and 2 do not present p values for the different groups. Are any of the differences statistically significant?

In Figure 3 there is a problem regarding the presentation of percentages for total sample.

Discussion: In this section you should discuss the results of your study comparing them with the results of previous studies. Only the results of other relevant studies are being presented.

Strengths and limitations sections: no causal relationship can be described since this is a cross-sectional study. In fact this is a limitation of your study. Please present and comment on other limitations (e.g. self-report of height and weight etc) of your study and not on the difficulties faced.

Finally, the manuscript would be improved by an editing by a native english speaker.

Author Response

(The authors gave the same response as above.)

Round 2

Reviewer 1 Report

The authors significantly improve their manuscipt

Author Response

Dear Reviewer,

Thank you for taking the time to read the submitted manuscript in the second round of review.

We have made additional improvements according to the comments from the 2nd Reviewer

Thanks again for your positive review and we wish you all the best!

Best Regards, Authors

Reviewer 2 Report

Dear authors thank you for the revised manuscript which is substantially improved. I suggest that you do not use the word phenomenon for orthorexia throughout the manuscript. Instead you may use the term condition.

In the tables please add the p values, even if there are non-significant differences.

in the discussion please present first the results of your own study and then compare them with the results of similar studies.

Author Response

Dear Reviewer,
At the outset, thank you for reviewing the manuscript. All suggestions indicated in the articel have been revised and applied. The red color indicates the changes made. Thank you very much for your review.

Best regards, Authors